# Prevalence of Patients Receiving Urate-Lowering Medicine in Greenland and Denmark: A Cross-Sectional Case–Control Study

**DOI:** 10.3390/ijerph19127247

**Published:** 2022-06-13

**Authors:** Sidsel Dan Hull, Marianne Welzel Andersen, Jessica Bengtsson, Nils Skovgaard, Marie Balslev Backe, Michael Lynge Pedersen

**Affiliations:** 1Department of Public Heath, University of Copenhagen, 1353 Copenhagen, Denmark; sidselhull@gmail.com (S.D.H.); jebe@sund.ku.dk (J.B.); 2Queen Ingrid’s Hospital, Nuuk 3900, Greenland; marianne.w.a@hotmail.com (M.W.A.); nsko@peqqik.gl (N.S.); 3Steno Diabetes Center Greenland, Queen Ingrid’s Hospital, Nuuk 3900, Greenland; milp@peqqik.gl; 4Greenland Center for Health Research, Institute of Health and Nature, University of Greenland, Nuuk 3905, Greenland

**Keywords:** gout, allopurinol, febuxostat, Greenland, Denmark, prevalence

## Abstract

This study estimates the age- and sex-specific prevalence of patients receiving urate-lowering therapy (ULT) in Greenland and compares the results with estimates in Denmark. Characteristics of patients receiving ULT in Greenland were compared to age- and sex-matched controls. The study was designed as a cross-sectional case–control study based on nationwide data from medical and population registers in Greenland and Denmark. The prevalence of patients receiving ULT was significantly lower in Greenland (0.55%) compared to Denmark (1.40%) (*p* < 0.001). In both countries, the prevalence increased with age and was higher among men compared to women. In Greenland, patients receiving ULT were more often overweight, and more frequently received blood glucose-, lipid-, and blood pressure-lowering medicine including diuretics compared to age- and sex-matched controls. The prevalence of patients receiving ULT was significantly lower in Greenland compared to Denmark. Common life-style related risk factors for hyperuricemia and gout were observed frequently among ULT-treated patients compared to controls. Along with an increasing prevalence of lifestyle-related diseases in Greenland, the prevalence of patients receiving ULT may increase in the years to come. More focus on detection and management of hyperuricemia and gout in Greenland is warranted to improve quality of health care.

## 1. Introduction

Urate-lowering therapy (ULT) involves various of strategies to reduce urate levels, typically pharmacological agents that either limits the formation of uric acid (allopurinol and febuxostat) or increase urinary excretion of uric acid (uricosuric agents) [1]. Allopurinol is recommended as first-line therapy in the treatment of hyperuricemia, and for patients who do not tolerate nor respond to allopurinol, febuxostat, and uricosuric agents are used as alternatives or supplements [1]. The main indication for allopurinol is preventing gout flares. Yet, it can be used in other conditions and recently it is has been speculated if therapies that reduce uric acid may slow chronic kidney disease progression and reduce cardiovascular morbidity [2]. Hyperuricemia has many causes, such as purine-rich food, alcohol, increasing age, cell lysis in cancer disease and several drugs, including diuretics and is associated with increased risk of gout [3]. A high level of hyperuricemia over a period of time can result in deposits of monosodium urate crystals in and around the joints and, thus, deformity and chronic usage-related pain, also referred to as gout. Globally, gout is the most common types of inflammatory arthritis [3]. The golden standard to confirm the diagnosis of gout includes identification of crystals in the joint fluid using polarizing microscopy [4]. However, in situations with limited access to microscopy, a combination of clinical features and measurements of high serum urate can be used to establish a gout diagnosis [1,4]. In Greenland, no access to polarized microscopy is available in the health care system. Furthermore, no physician within the field of rheumatology is employed in Greenland. Thus, diagnosis and management of gout in Greenland is performed in a primary care setting context and has to be based on clinical features and elevated serum urate levels. The only available ULT in Greenland is allopurinol, and exclusively prescribed for patients with gout and hyperuricemia. Patients experiencing acute gout flares are treated with colchicine or nonsteroidal anti-inflammatory drugs (NSAID), and thus not all gout patients receive prophylactic ULT, such as allopurinol. In general, even though ULT is the first-line therapy for gout and hyperuricemia, only a third of gout patients are treated with ULT, and of these only 50% adhere to the treatment [3]. In addition to medical treatment, recent international recommendations on gout management emphasized the need to search for risk factors for hyperuricemia and comorbidities, including obesity, chronic kidney disease, cardio metabolic diseases, medications (diuretics, low-dose aspirin, cyclosporine, tacrolimus), and diet [3]. During recent decades, the society and living conditions in Greenland have changed dramatically, resulting in an increasing prevalence of obesity and lifestyle-related diseases, such as cardiovascular disease and diabetes [5,6,7]. Likewise, the prevalence of patients with indication for UTL in Greenland may be rising, yet, unexplored. Former studies have revealed a high proportion (70–80%) of undiagnosed diabetes in Greenland [8,9]. This knowledge has been addressed by the healthcare system in Greenland and today the proportion of undiagnosed diabetes is low (approximately 24%) in a global perspective [7]. Similarly, it can be hypothesized that treatment with UTL has gained limited attention in Greenland and quality of care may be optimized.

The objective of this study was to estimate the age- and sex-specific prevalence of patients receiving ULT in Greenland and compare these results with estimates in Denmark. Furthermore, we aimed to explore common characteristics for patients receiving ULT in Greenland compared to age- and sex-matched controls not receiving ULT.

## 2. Material and Methods

### 2.1. Study Design

The study was designed as a cross-sectional case–control study based on nationwide data from medical and population registers in Greenland and Denmark.

### 2.2. Setting

Greenland is the largest island in the world covering an area of more than 2 million km^2^. Despite its size, the country is sparsely populated with about 56,000 people living in towns and small settlements widely spread along the coastline. Greenland is located in the arctic region and is a country within the Danish Realm with self-rule since 2009 [6]. The healthcare system in Greenland is divided into five healthcare regions, each covering a number of towns and settlements. A regional hospital is located in the largest town within each region. The remaining towns have a primary healthcare centre, and the small settlements have smaller healthcare units. Specialized and secondary healthcare are delivered at Queen Ingrid’s Hospital in the capitol, Nuuk [6].

Denmark is a country located in Scandinavia that covers an area of about 43,000 km^2^ and has a population of 5.8 million people. The healthcare system in Denmark consists of a primary and secondary sector. The primary sector includes general practitioners with the authority to refer patients to the secondary sector, which includes hospitals and other specialized medical units, such as private specialists [10]. 

Healthcare services are free of charge in both countries. In Greenland, prescribed medicine is also free of charge, whereas it requires a payment from the patient in Denmark [10]. 

### 2.3. Study Population

In Greenland, the only ULT used is allopurinol. All patients aged ≥ 20 years with an active drug prescription for allopurinol on 20 April 2021 in Greenland were identified using the Anatomic Therapeutic Chemical (ATC) code M04AA01 through the electronic medical record (EMR) (N = 216). Age- and sex-matched controls were randomly identified on the same date in the EMR with a ratio of 1:1 (N = 216). The control group did not include any patients treated with allopurinol. No patients nor controls from Tasiilaq district (less than 10% of the population in Greenland) were included, since the healthcare system in this area uses a different EMR system than the rest of Greenland.

In Denmark, all patients aged ≥ 20 years with a purchase of either allopurinol and/or febuxostat were identified using the ATC code M04AA through the online medical register Medstat (N = 63,190) [11]. The latest available data included all patients with a purchase of allopurinol and/or febuxostat in the period 1 January 2019 to 31 December 2019. 

No information on the dose of ULT was available for any of the registers.

The background population in Greenland was extracted from Statistics Greenland [12] and included all people aged ≥ 20 years living in Greenland (except Tasiilaq) on 1 January 2021 (N = 39,325). In Denmark, the background population was extracted from Statistics Denmark and included the total population aged ≥ 20 years living in Denmark on 1 January 2019 (N = 4,504,196) [13].

### 2.4. Statistical Analyses

#### 2.4.1. Covariates

Information on age, sex, daily smoking (yes/no), physical inactivity (<5000 footsteps per day), blood pressure (mmHg), level of uric acid (mmol/L), estimated glomerular filtration rate (eGFR [mL/min]), and glycated haemoglobin (HbA1c [mmol/mol]) was extracted from the lifestyle table and the lab card in the EMR. Additionally, body mass index (BMI) was calculated using information on weight and height (kg/m^2^). If available in the EMR, blood pressure measured at home was used. Otherwise, blood pressure measured at the health facility was used. HbA1c, uric acid, and eGFR were measured from venous blood samples. Analysis of HbA1c level was performed on a Tosoh G8 HPLC analyser, while uric acid level and eGFR were analysed with an Architect 8000T from Abbot^®^. Only the most recent measurements (<2 years from the date of the data extraction on 20 April 2021) for each individual were included in the analyses. 

Medicine use was assessed through present drug prescriptions extracted from the EMR on 21 April 2021. Blood glucose-lowering medicine was defined as drugs with ATC code A10. Lipid-lowering medicine included ATC code C10, and finally, blood pressure-lowering medicine was defined as drugs with ATC codes C02–C03 and C07–C09, where C03 covers diuretics, a sub-type of blood pressure lowering medicine. 

#### 2.4.2. Statistics 

Prevalence estimates were calculated using the background population as denominator. We calculated prevalence ratios (PR) to compare prevalence estimates and used Chi-square tests to compare frequencies between groups. Covariates were described with medians and interquartile ranges (IQR). Mann U tests were used to compare medians. All estimates were calculated with 95% confidence intervals (CI) and *p*-values below 0.05 were considered significant. All statistical analyses were performed in IMB SPSS Statistics 27. 

#### 2.4.3. Ethics

The study was approved by the Ethics Committee for Scientific Research in Greenland (reference no. 2021-1484) and by the Agency for Health and Prevention in Greenland.

## 3. Results

Among the 216 patients receiving ULT in Greenland, 79% (171) were men. Among the 63,190 patients receiving ULT in Denmark, 78% (49,419) were men. In Denmark, 99% of patients receiving ULT were treated with allopurinol. 

### 3.1. Prevalence Estimates

The age- and sex-specific prevalence of patients receiving ULT in Greenland and Denmark is shown in Table 1. The prevalence of patients receiving ULT increased with age for both men and women in both countries. In Greenland, the prevalence of patients receiving ULT was higher among men compared with women across all age groups, although the difference was not significant in the age group 20–39 years. A similar trend was seen in Denmark, where the prevalence was significantly higher (*p* < 0.001) among men compared to women in all age groups.

The overall prevalence of patients receiving ULT in Greenland was 0.55% (0.24% among women and 0.82% among men) whereas in Denmark, the overall prevalence was 1.40% (0.60% among women and 2.22% among men) and significantly higher than in Greenland (*p* < 0.001).

Table 1 show that in the age groups 20–79 years, the prevalence of ULT prescriptions was lower among Greenlandic men (overall prevalence: 0.82%, 95% CI: 0.69;0.94) compared to Danish men (overall prevalence: 2.22%, 95% CI: 2.20;2.24). In addition, the prevalence of ULT prescriptions was lower among Greenlandic women within the age group 60–79 years (0.62%, 95% CI: 0.36;0.88) compared to Danish women (1.16%, 95% CI 1.14;1.19). For women in the remaining age groups, the differences between Greenland and Denmark were not significant.

### 3.2. Characteristics of Patients Receiving ULT in Greenland

The characteristics of patients receiving ULT in Greenland are illustrated in Table 2. Men had a higher level of uric acid and a higher eGFR compared to women, while a higher proportion of women smoked cigarettes on daily basis compared to men.

Patients receiving ULT had a higher BMI and were more often overweight compared to age- and sex-matched controls. In addition, patients receiving ULT had a higher level of uric acid and a lower eGFR compared to the control group. Patients receiving ULT also had a significantly higher use of medicine compared to the control group. Among patients receiving ULT in Greenland, 20.8% (45 patients) received blood glucose-lowering medicine, 38.9% (84 patients) received lipid-lowering medicine, and 73.6% (159 patients) received blood pressure-lowering medicine among whom 45.8% (99 patients) received diuretics. 

## 4. Discussion

Our study showed a significantly lower overall prevalence of patients receiving ULT in Greenland (0.55%) compared to Denmark (1.40%). In both countries, the prevalence increased with age and was higher among men compared to women across all age groups. In Greenland, patients receiving ULT had a higher BMI, were more often overweight, had a higher level of uric acid, a lower eGFR and more frequently received blood glucose-lowering medicine, lipid-lowering medicine, as well as blood-pressure-lowering medicine including diuretics compared to age- and sex-matched controls. 

### 4.1. Prevalence

We found the prevalence of patients receiving ULT to be 0.55% in Greenland and 1.40% in Denmark. This is lower than reported in a recent population based study among older (65 years or above) people in Poland [14]. They reported that 2.8% of men and 2.2% of women were receiving ULT. However, compared to the older age group in this study, the prevalence of patients using ULT is actually higher in Denmark (4.82% among men aged 60–79 years old) and quite comparable to Greenland (2.34% and 5.34% among men in age group 60–79 years and 80 years or above, respectively). A recent study from New Zealand reported an overall prevalence of patients receiving ULT at around 3%, higher among men than among women and higher among Maori than non-Maori [15]. Potentially, several factors may influence the prevalence of patients using ULT, among those, in particular with access to high-quality health care and variation in the true prevalence of gout in different populations. Globally, the prevalence of gout has been estimated to vary hugely from <1% to 6.8% [3], partially explained by variation in research methods. Thus, it has been reported that prevalence of gout based on administrative data including diagnosis in the EMR has limited validity [16]. 

The reported prevalence of treatment with ULT in Greenland is lower than former prevalence estimates of gout [17,18]. Thus, the age-standardized prevalence of gout in Greenland based on statistical modelling has been reported to be 0.96% (95% CI: 0.85–1.09) [18]. Since only around a third of patients with gout [3] are actually treated with ULT, the current use in Greenland suggests that the prevalence of gout may have increased (three times 0.55% vs. 0.96%) compared to the former study. A Danish study has reported the prevalence of gout in Denmark to be 0.68% (95% CI 0.68;0.69) based on data from the Danish National Patient Register (DNPR) [17]. However, the DNPR only includes data from hospital records and a few private practices, leaving out milder cases of gout managed in primary care. The higher prevalence of use ULT in our study (1.40% compared to 0.68%) most likely reflect that some patients are treated with ULT in the primary care setting without referral to a rheumatologist. In addition, in Denmark, ULT may be used for other conditions included urate sediments in the urine. However, this is not a common condition compared to gout and are expected to represent only a minimal part of the patients treated with ULT in this study. 

The observed lower prevalence of patients receiving ULT in Greenland compared to Denmark is in line with previous studies investigating the prevalence of patients treated with antidiabetics and anti-depressive medicine also finding a lower prevalence in Greenland compared to Denmark [10,19]. The difference between Greenland and Denmark might be due to an actual lower prevalence of patients with gout in Greenland compared to Denmark. However, differences in healthcare access and awareness of gout in the two countries may also be an important contributing factor. In general, lower socio-economic health status is observed in Greenland compared to Denmark concerning education level, mean income, daily smoking, and life expectancy, which may influence the health-seeking behavior and awareness of hyperuricemia in the health care system in Greenland.

### 4.2. Characteristics

We found a higher prevalence of patients receiving ULT among men compared to women, and that the prevalence increased with age, which was expected as ULT is primary used for gout, a disease observed more frequently among men and older people [3,20]. In addition, patients receiving ULT in Greenland had a higher level of uric acid and a lower eGFR compared to the age- and sex-matched controls. This was also expected since hyperuricemia can lead to nephropathy and the fact that nephropathy on its own leads to hyperuricemia due to decreased renal excretion of urate.

Our study also showed a significant association between patients receiving ULT and obesity, which we expected since obesity is a well-known risk factor for gout [3]. The result is supported by a meta-analysis from 2018 showing a more than two-times higher risk of developing gout among obese individuals (≥30 kg/m^2^) compared to individuals with a BMI < 30 kg/m^2^ [21].

Regarding use of medicine, we observed that a higher proportion of patients receiving ULT received blood glucose-lowering medicine, lipid-lowering medicine, as well as blood-pressure-lowering medicine and diuretics compared to the age- and sex-matched controls indication higher prevalence of diabetes, hypertension, and dyslipidaemia in the ULT receiving group. This is in line with former studies reporting diabetes as a comorbidity to gout (and vice versa) [3,22]. Additionally, the higher use of lipid-lowering medicine is in accordance with the results of a large case–control study showing an association between gout and cardiovascular disease and hyperlipidaemia [23]. We observed that a greater proportion of patients receiving ULT in Greenland received diuretics compared to the control group, which is in accordance with a previous cohort study investigating the impact of diuretics on gout reporting a relative risk of 1.77 (95% CI 1.42;2.20) for men taking diuretics compared to men who did not take diuretics [24]. In general, blood pressure-lowering medicine and diuretics have previously been reported as risk factors for gout [21,24]. Well-known risk factors for gout, such as high age, male sex, treatment with diuretics, and indicators for diabetes, hypertension, and dyslipidaemia were frequent among patients receiving ULT, indicating that the risk factors for gout in Greenland may be the same as elsewhere. 

### 4.3. Strengths and Limitations

This was the first study to estimate the prevalence of patients receiving ULT in Greenland. A major strength was the use of nationwide data for both Greenland and Denmark including the majority (96%) of the Greenlandic population aged ≥ 20 years and the entire Danish population ≥ 20 years, and thereby minimizing the risk of information- and selection bias. 

Limitations include the relatively low number of patients in some age groups, primarily among women, which limited the statistical power within these groups. The statistical comparison between patients receiving ULT and the control group is additionally limited by missing information on BMI, blood pressure, HbA1c, uric acid and eGFR, especially in the control group, which may reflect that the control group has less healthcare contact than the group of patients receiving ULT. The observed differences and the characteristics of patients receiving ULT should, therefore, be interpreted with some consideration. 

The comparison of prevalence estimates between Greenland and Denmark should also be made with clear reservations. In Greenland, patients receiving ULT were identified through the EMR whereas in Denmark, patients receiving ULT were identified through purchases of allopurinol and/or febuxostat. Consequently, the prevalence estimates in Greenland and Denmark are not fully comparable. Additionally, the period observed in Greenland (2021) and Denmark (2019) was not identical. However, we included the most recent data available in the two countries.

## 5. Conclusions

We found a significantly lower overall prevalence of patients receiving ULT in Greenland (0.55%) compared to Denmark (1.40%) (*p* < 0.001). Characteristics, such as high age, male sex, use of diuretics, blood-glucose-lowering medicine, lipid-lowering medicine, as well as blood-pressure-lowering medicine were more frequent among patients receiving ULT, indicating that the risk factors for hyperuricemia and gout in Greenland are the same as elsewhere. Along with the increasing prevalence of these conditions in Greenland, the need for ULT may increase in the near future. Focus on detection and management of hyperuricemia and gout in Greenland is warranted.

## Figures and Tables

**Table 1 ijerph-19-07247-t001:** Age- and sex-specific prevalence of patients receiving ULT in Greenland and Denmark.

	Total Prevalence (95% CI) n/N			Women Prevalence (95% CI) n/N			Men Prevalence (95% CI) n/N			Men vs. Women	
										Greenland	Denmark
Age (Years)	Greenland	Denmark	PR (95% CI) *p*-Value	Greenland	Denmark	PR (95% CI) *p*-Value	Greenland	Denmark	PR (95% CI) *p*-Value	PR (95% CI) *p*-Value	PR (95% CI) *p*-Value
20–39	0.06 (0.02;0.10) 10/16,368	0.13 (0.13;0.14) 1947/1,465,040	0.46 (0.25;0.86) **0.012**	0.04 (0.00;0.08) 3/7961	0.06 (0.05;0.06) 423/719,058	0.64 (0.21;1.99) 0.438	0.08 (0.02;0.14) 7/8407	0.20 (0.19;0.21) 1524/745,982	0.41 (0.19;0.86) **0.014**	2.21 (0.57;8.55) 0.238	3.47 (3.12;3.87) **<0.001**
40–59	0.45 (0.34;0.55) 65/14,604	0.88 (0.87;0.90) 13,799/1,560,423	0.50 (0.39;0.64) **<0.001**	0.25 (0.13;0.38) 17/6678	0.21 (0.20;0.22) 1659/776,908	1.19 (0.74;1.92) 0.470	0.61 (0.43;0.78) 48/7926	1.55 (1.52;1.58) 12,140/783,515	0.39 (0.29;0.52) **<0.001**	2.38 (1.37;4.14) **0.001**	7.26 (6.89;7.64) **<0.001**
60–79	1.59 (1.31;1.87) 123/7749	2.93 (2.90;2.96) 35,648/1,214,987	0.54 (0.45;0.65) **<0.001**	0.62 (0.36;0.88) 21/3389	1.16 (1.14;1.19) 7280/626,615	0.53 (0.35;0.82) **0.003**	2.34 (1.89;2.79) 102/4360	4.82 (4.77;4.88) 28,368/588,372	0.49 (0.40;0.59) **<0.001**	3.78 (2.36;6.05) **<0.001**	4.15 (4.04;4.26) **<0.001**
≥80	2.98 (1.62;4.34) 18/604	4.47 (4.39;4.55) 11,796/263,746	0.67 (0.42;1.07) 0.076	1.15 (0.03;2.27) 4/348	2.76 (2.68;2.84) 4409/159,926	0.42 (0.16;1.12) 0.067	5.47 (2.68;8.25) 14/256	7.12 (6.96;7.27) 7387/103,820	0.77 (0.45;1.32) 0.306	4.76 (1.55;14.62) **0.002**	2.58 (2.48;2.68) **<0.001**
Overall	0.55 (0.48;0.62) 216/39,325	1.40 (1.39;1.41) 63,190/4,504,196	0.39 (0.34;0.45) **<0.001**	0.24 (0.17;0.32) 45/18,376	0.60 (0.59;0.61) 13,771/2,282,507	0.41 (0.30;0.54) **<0.001**	0.82 (0.69;0.94) 171/20949	2.22 (2.20;2.24) 49,419/2,221,689	0.37 (0.32;0.43) **<0.001**	3.33 (2.40;4.63) **<0.001**	3.69 (3.62;3.76) **<0.001**

95% CI: 95% Confidence interval, n: Number of patients treated with allopurinol in Greenland and patients treated with allopurinol or Febuxostat in Denmark, N: Number of people in the background population. *p*-values below 0.05 are in bold. PR = Prevalence Ratio.

**Table 2 ijerph-19-07247-t002:** Characteristics of patients treated with allopurinol in Greenland (women and men) and a control group matched on age and sex.

Characteristic	Women		Men			Total		Control Group		
	Median (IQR)	N (45)	Median (IQR)	N (171)	*p*	Median (IQR)	N (216)	Median (IQR)	N (216)	*p*
Age (years)	62 (19.50)	45	64 (16.00)	171	0.497	64 (16.75)	216	64 (16.75)	216	-
BMI (kg/m^2^)	33 (10.08)	23	31 (7.85)	97	0.736	32 (8.29)	120	29 (6.48)	64	**<0.001**
Blood pressure (mmHg)										
Systolic	140 (17.00)	31	135 (18.25)	122	0.069	136 (17.00)	153	140 (23.50)	82	0.182
Diastolic	83 (16.00)	31	81 (10.25)	122	0.17	81 (12.00)	153	82 (13.25)	82	0.594
HbA1c (mmol/mol)	42 (8.00)	42	43 (9.00)	150	0.869	42 (8.00)	192	42 (5.25)	118	0.464
Urid acid level (mmol/L)	0.38 (0.17)	41	0.43 (0.18)	133	**0.041**	0.42 (0.19)	174	0.34 (0.14)	25	**0.023**
eGFR (mL/min)	65 (42.00)	39	73 (31.00)	143	**0.01**	71 (33.00)	182	81 (22.00)	123	**<0.001**
	% (n)	N	% (n)	N	*p*	% (n)	N	% (n)	N	*p*
Daily smoking	45.5 (10)	22	23.7 (23)	97	**0.04**	27.7 (33)	119	40.9 (27)	66	0.067
Overweight (BMI ≥ 30)	65.2 (15)	23	66.0 (64)	97	0.945	65.8 (79)	120	46.9 (30)	64	**0.013**
Physical inactive	66.7 (6)	9	52.5 (32)	61	0.424	54.3 (38)	70	54.5 (24)	44	0.978
Glucose-lowering medicine	17.8 (8)	45	21.6 (37)	171	0.571	20.8 (45)	216	6.9 (15)	216	**<0.001**
Lipid-lowering medicine	33.3 (15)	45	40.4 (69)	171	0.39	38.9 (84)	216	19.0 (41)	216	**<0.001**
Antihypertensive medicine	68.9 (31)	45	74.9 (128)	171	0.419	73.6 (159)	216	38.4 (83)	216	**<0.001**
Diuretics	51.1 (23)	45	44.4 (76)	171	0.425	45.8 (99)	216	14.8 (32)	216	**<0.001**

HbA1c: glycated haemoglobin. eGFR: estimated glomerular filtration rate. Physical inactive is defined as <5000 footsteps per day. % refers to the percentage of patients with available data.

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
