# Peer review of "Prevalence of Patients Receiving Urate-Lowering Medicine in Greenland and Denmark: A Cross-Sectional Case–Control Study"

_ijerph, 2022, doi:10.3390/ijerph19127247_

Round 1
Reviewer 1 Report
This is an interesting study assessing the prevalence of ULT in Greenland. This is the first study to assess the prevalence of ULT in this population. For Greenland population, authors used electronic medical records.
Firstly, thanks to the authors for providing us with elements to better understand the functioning of the health care system in Greenland.
In this study, the authors compare the prevalence of Greenland patients under ULT with the prevalence of patients under ULT in Denmark. A lower prevalence was found in the Greenland population (0.55%) compared to Denmark. The authors found the factors "classically" associated with gout, namely high plasmatic uric acid level, overweight, dyslipidemia, impaired renal function and a male predominance.
The results presented are in line with the known knowledge in the field of gout.
However, I would have several remarks, notably on the methodology:
1- For the Greenland population, there is no mention of treatment with febuxostat. Were no patients taking this treatment or was this not recorded in the electronic patient records?
2- Patients from the Tassilaq region were excluded because of the use of a different electronic system. Wouldn't it be interesting to include this population in this study, which would allow the whole population of Greenland to be taken into account, even if the number of patients is small?
3- As the authors mention, one of the limitations of this study is the use of different databases on two populations, making comparison difficult, especially as the mode of evaluation is different.
A- In the Greenland population, the prescription of allopurinol is retained but not its continuation. Why this difference?
B- It would be more accurate to compare the 2 populations:
1- over the same period (2021 greenland, 2019 danish)
2- on the same criteria (regular use of ULT)
Author Response
Response to Reviewer 1 in bold after each point raised.
Reviewer I
This is an interesting study assessing the prevalence of ULT in Greenland. This is the first study to assess the prevalence of ULT in this population. For Greenland population, authors used electronic medical records.
Firstly, thanks to the authors for providing us with elements to better understand the functioning of the health care system in Greenland.
In this study, the authors compare the prevalence of Greenland patients under ULT with the prevalence of patients under ULT in Denmark. A lower prevalence was found in the Greenland population (0.55%) compared to Denmark. The authors found the factors "classically" associated with gout, namely high plasmatic uric acid level, overweight, dyslipidemia, impaired renal function and a male predominance.
The results presented are in line with the known knowledge in the field of gout.
Thank you very much for acknowledging our work and for your time to provide valuable suggestions.
However, I would have several remarks, notably on the methodology:
1- For the Greenland population, there is no mention of treatment with febuxostat. Were no patients taking this treatment or was this not recorded in the electronic patient records?
Good point. In Greenland, prescribed medicine is free of charge for all patients. At the same time, only national recommend drugs can be prescribed. Concerning ULT, only allopurinol is recommend in Greenland and no patients receive febuxostat. In the method section, we added the following sentence: “In Greenland, the only ULT used is allopurinol.”
2- Patients from the Tassilaq region were excluded because of the use of a different electronic system. Wouldn't it be interesting to include this population in this study, which would allow the whole population of Greenland to be taken into account, even if the number of patients is small?
This is a very good point and we would really like to include the remaining part of Greenland too. However, it was not possible to obtain the information from this district.
3- As the authors mention, one of the limitations of this study is the use of different databases on two populations, making comparison difficult, especially as the mode of evaluation is different.
A- In the Greenland population, the prescription of allopurinol is retained but not its continuation. Why this difference?
Thank you for pointing this out. Actually, we included all patients with an active prescription in the EMR including continuations reflecting the whole population regular treated with ULT. We have clarified this in the methods section:”……. with an active drug prescription for allopurinol on…………”.
B- It would be more accurate to compare the 2 populations:
1- over the same period (2021 greenland, 2019 danish)
Agree, we would have preferred to do so. However, we included the most recent data available in both countries to obtain the most valid data. Until recently, manually recorded prescriptions have been used in some parts of Greenland and using older data from (2019) in Greenland would consequently lead to an underestimation of the true population using ULT. We have added the following in the limitation section. “Also, the period observed in Greenland (2021) and Denmark (2019) was not identical. However, we included the most recent data available in the two countries.”
2- on the same criteria (regular use of ULT)
Agree, as mentioned above and clarified in the method section, we included patients with regular use of ULT in both Denmark and Greenland.
Reviewer 2 Report
This paper aims to answer several scientific questions, i.e. prevalence of patients receiving urate lowering therapy (ULT) in Greenland vs. Denmark, with age and sex-specific subgroup analyses. Patients who received ULT were also clinically characterized to know if there are differences between ULT-receiving patients in Greenland vs. elsewhere.
Very well written article. Objectives were clearly stated, methods being used were appropriate. Data are presented clearly and very easily understood. Conclusions were appropriately drawn, based on available data. Limitations of the study are clearly stated.
Major comments:
- Are there any differences in socio-economic status between Greenlandic and Danish patients? This might also correlate with access to information, or awareness of disease, as indicated in 4.1.
- Could the authors comment on the dose of ULT received by the patients?
Minor comment
A typographical error: “Similarly, it can be hypnotized that treatment with UTL has gained limited attention in Greenland and quality of care may be optimized”, I believe it is meant to read “hypothesized” not "hypnotized".
Author Response
Response to Reviewer 2 in bold after each point raised.
Comments and Suggestions for Authors
This paper aims to answer several scientific questions, i.e. prevalence of patients receiving urate lowering therapy (ULT) in Greenland vs. Denmark, with age and sex-specific subgroup analyses. Patients who received ULT were also clinically characterized to know if there are differences between ULT-receiving patients in Greenland vs. elsewhere.
Very well written article. Objectives were clearly stated, methods being used were appropriate. Data are presented clearly and very easily understood. Conclusions were appropriately drawn, based on available data. Limitations of the study are clearly stated.
Thank you very much for your evaluation.
Major comments:
- Are there any differences in socio-economic status between Greenlandic and Danish patients? This might also correlate with access to information, or awareness of disease, as indicated in 4.1.
Good point. We have included the following sentence in the section 4.1: “………lower socio-economic-health status is observed in Greenland compared to Denmark concerning education level, mean income, daily smoking, and life expectancy, which may influence the health seeking behavior and awareness…...”
- Could the authors comment on the dose of ULT received by the patients?
Unfortunately, no information on dose were available in the two registers. A comment on this has been added in the methods section. “No information on the dose of ULT was available for any of the registers.”
Minor comment
A typographical error: “Similarly, it can be hypnotized that treatment with UTL has gained limited attention in Greenland and quality of care may be optimized”, I believe it is meant to read “hypothesized” not "hypnotized".
Certainly, thank you for noticing, corrected in the text.